# Species and Trait-Based Reconstructions of the Hydrological Regime in a Tropical Peatland (Central Sumatra, Indonesia) during the Holocene Using Testate Amoebae

Andrey N. Tsyganov [1,*,†], Elena A. Malysheva [2], Yuri A. Mazei [1,3,4], K. Anggi Hapsari [5,6], Hermann Behling [5], Supiandi Sabiham [7], Siria Biagioni [5] and Valentyna Krashevska [8,*,†]

1 Department of General Ecology and Hydrobiology, Lomonosov Moscow State University, Leninskie Gory 1, 119991 Moscow, Russia
2 Department of General Biology and Biochemistry, Penza State University, Krasnaya Str. 40, 440036 Penza, Russia
3 A.N. Severtsov Institute of Ecology and Evolution, Russian Academy of Sciences, Leninsky Av. 33, 119071 Moscow, Russia
4 Faculty of Biology, Shenzhen MSU-BIT University, Shenzhen 518100, China
5 Department of Palynology and Climate Dynamics, Albrecht-von-Haller-Institute for Plant Sciences, University of Göttingen, 37073 Göttingen, Germany
6 College of Life and Environmental Sciences, University of Exeter, Exeter EX4 4RJ, UK
7 Department of Soil Science and Land Resource, Bogor Agricultural University (IPB), Bogor 16162, Indonesia
8 J.F. Blumenbach Institute of Zoology and Anthropology, University of Göttingen, 37073 Göttingen, Germany
* Correspondence: andrey.tsyganov@bk.ru (A.N.T.); valentyna.krashevska@biologie.uni-goettingen.de (V.K.); Tel.: +79-063-973-238 (A.N.T.); +49-055-139-254-45 (V.K.)
† These authors contributed equally to this work.

**Abstract:** Paleoecological reconstructions of hydrological regimes in tropical peatlands during the Holocene are important for the estimation of their responses to changing environments. However, the application of some widely used proxies, such as testate amoebae, is hampered by poor knowledge of their morphology and ecological preferences in the region. The aim of this study is to describe the morphospecies composition of sub-fossil testate amoebae in deposits of a tropical peatland in Central Sumatra (Indonesia) during the Holocene and reconstruct the hydrological regime using morphospecies- and functional-trait-based approaches. In total, 48 testate amoeba morphospecies were observed. Based on morphospecies composition, we distinguished three main periods of peatland development (13,400–8000, 8000–2000, 2000 cal yr BP–present). The application of the morphospecies-based transfer function provided a more reliable reconstruction of the water regime in comparison to the functional trait-based one. The weak performance of the latter might be related to the poor preservation of shells and the greater variation in the functional traits in sub-fossil communities as compared to the training set and linear modeling approach. These results call for future studies on the functional and morphospecies composition of testate amoebae in a wider range of tropical peatlands to improve the quality of hydrological reconstructions.

**Keywords:** transfer function; water table depth; protists; mires; Sungai Buluh peatland; weighted-averaging; multiple regression

## 1. Introduction

Peatland ecosystems are regarded as highly important by both ecologists and climate scientists due to their essential role in the regulation of the local and regional water balance and their high capacity for carbon storage [1]. Recent studies indicate that tropical peatlands cover 23–30% of the total peatland area in the world, i.e., 90 to 170 Mha [2–4], which is much greater than the previous estimates of 36–44 Mha [5–7]. This substantial increase in the tropical peatland area is related to new discoveries of large peatlands in remote locations with low accessibility [8,9]. While high- and mid-latitude peatlands and peat-forming ecosystems have been investigated extensively, tropical peatlands remain poorly studied, especially with respect to their origin, development, main environmental drivers, links to climatic forces, and resilience to anthropogenic impacts [10,11]. This substantial area (21 Mha) of tropical peatlands is located in Southeast Asia [2], mainly in Indonesia (13.43 Mha), where they generally cover the coastal regions of the islands Sumatra and Borneo [5]. Available paleoecological studies [10,12] have shown the importance of sea-level change and rainfall in the formation and development of these peatlands and peat swamp forests. However, despite the fact the hydrological regime is one of the main factors affecting peatlands, which also determines the storage and flux of carbon in these ecosystems, it has been rarely reconstructed in paleoecological studies on tropical peatlands.

Among the proxies for the reconstruction of the hydrological regime in peatlands, peat humification, hydrogen isotopes [13], and testate amoebae have great potential in tropical settings [14–16]. Testate amoebae are a diverse polyphyletic group of free-living eukaryotic microbes that are characterized by the presence of an external shell [17]. They form species-rich and abundant communities in soils, mosses, and aquatic environments worldwide, being especially diverse in tropical regions [18–21]. The species composition of the testate amoeba communities in peatlands is strongly controlled by surface wetness (usually estimated as water table depth, WTD) [22]. Furthermore, testate amoeba shells are well preserved in peat deposits, which makes them a good proxy for the reconstruction of the hydrological regime. In recent years, a number of transfer functions have been developed to perform a quantitative reconstruction of WTD based on morphospecies composition of sub-fossil testate amoeba communities in peat, but their application has mostly been limited to high- and mid-latitude peatlands [23–26]. An application of this approach to tropical peatlands was hampered by poor knowledge of morphology and ecological preferences of testate amoebae in this region. Only a few studies used testate amoeba for paleoecological reconstructions or the development of transfer functions in the tropical peatlands, e.g., montane peatlands in Hawaii and Columbia [15,16], lowland Amazonian peatlands [27–29], and Indonesian peatlands [11,20]. However, recently, a functional trait-based approach has been developed in the application of testate amoebae that might help to overcome the limitations of the morphospecies-based transfer functions related to incomplete regional taxa inventories [30,31].

Functional traits have been defined as the key characteristics of an organism that determine its fitness or performance [32] and were originally used to gain a mechanistic understanding of key processes in community ecology [33]. This approach was later adopted for paleoecological studies on plant macrofossils and pollen and explored the response of life-history traits to climatic and environmental changes [34,35]. The trait-based approach appeared to be particularly useful in application to testate amoebae [36], in both contemporary [37,38] and paleoecological settings [30,39–43]. Two functional-trait-based transfer functions [20,31] were developed for the reconstruction of surface wetness in peatlands of the Southern Hemisphere, where data on species diversity of testate amoebae are particularly scanty. The main aim of this study is to describe the species composition of sub-fossil testate amoebae in peat deposits of a tropical peatland in Central Sumatra (Indonesia) during the Holocene and reconstruct the hydrological regime using species and functional trait approaches. Here, we use recently developed transfer functions [20] based on species and functional traits of modern testate amoebae in the Sungai Buluh peatland.

## 2. Materials and Methods

### 2.1. Study Region

The Sungai Buluh peatland is in the Jambi province, Sumatra, Indonesia (Figure 1a). Harboring around 700,000 ha of peatland, Jambi is currently the province with the third largest peatland area in Sumatra [5]. The region is characterized by a tropical-humid climate with a mean annual temperature of around 26–27 °C and a mean annual precipitation of 2400 mm [44]. Throughout the year, temperature varies insignificantly, and the drier season generally lasts from May to September [44]. The precipitation patterns of the region are influenced by the Asian–Australian monsoon and the Intertropical Convergence Zone (ITZC; [45]), while its interannual rainfall variability is controlled by El Niño Southern Oscillation (ENSO; [44]) and the Indian Ocean Dipole (IOD; [45]). The main river of Jambi is The Batang Hari, which flows throughout the province.

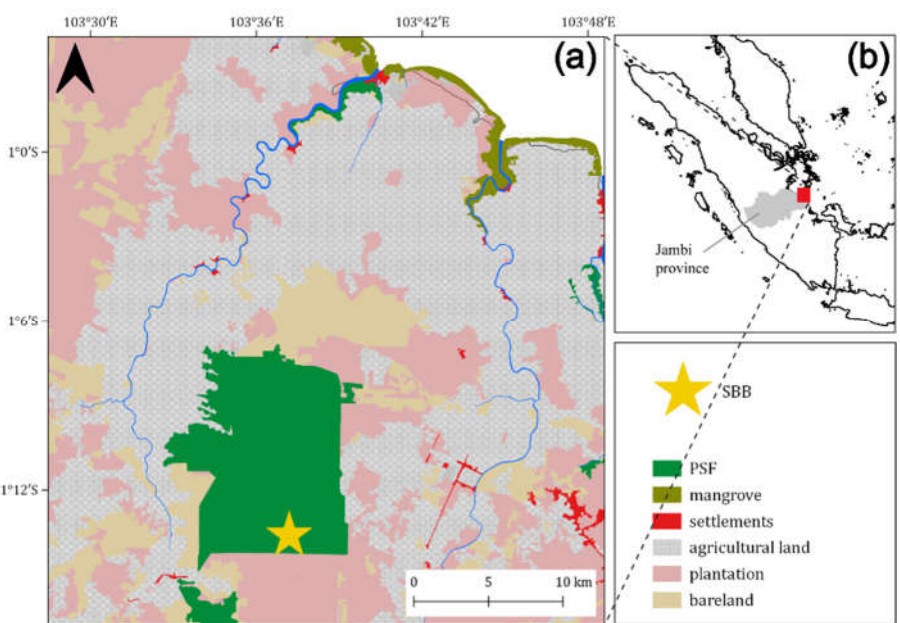

**Figure 1.** Map of the study region (**a**) and the location of the study site (**b**). SBB—Sungai Buluh peatland, PSF—peat swamp forest.

### 2.2. Study Site

The study site (Sungai Buluh peatland) is a secondary peat swamp forest (area 18,000 ha) located approximately 30 km northeast of the city of Jambi and 19 km from the coastline (Figure 1b). From the geomorphological and hydrological point of view, the peatland of Sungai Buluh is an extensive coastal peat dome, which is delimited by two rivers located to the east and west sides. The present tree cover is generally formed by *Shorea pauciflora*, *Dyera polyphylla,* and *Gonystylus bancanus* and has been regenerating since the selective logging activities in the 1960s and 1970s and enrichment planting in 2003 [46]. The ENSO-related fires also reduced the peat swamp forest canopy of Sungai Buluh in 1997 [47]. The area surrounding the Sungai Buluh peatland has been converted to agricultural fields and plantations (e.g., *Acacia* pulp woods and oil palm [48]). The water table in the peatland fluctuates from 0.0 to 0.7 m (with a mean value of 0.3 m) below the peat surface [49]. The Sungai Buluh peatland is located approximately 28 km north of the remnant of the ancient kingdom in Sumatra, the Malayu Empire. The temple complex of Muara Jambi covers around 12 km² and dates to the period of the 9–14th centuries. The complex was located on the banks of the Batang Hari River and served as the capital of the Melayu Empire [50,51]. The deposits of the Sungai Buluh have been previously used for studying the dynamics of environmental and peat carbon accumulation [49] and peatland resilience to anthropogenic disturbances [52].

### 2.3. Peat Sampling

Peat deposits (SB-B peat core, a total length of 350 cm) from the Sungai Buluh peatland (1.236111°S 103.62°E; 18 m a.s.l.) were extracted with a closed D-chamber corer (a diameter of 5 cm) in 2013 [53]. The deposits were described following the Southeast Asian peat classification [54] and comprise a peat layer (0–240 cm) and underlying clay (240–350 cm) [49]. To determine the age of the deposits, eight samples of the peat core (including bulk sediment, organic bulk sediment, organic samples, charred particles, and seed material) were collected for Accelerated Mass Spectrometer radiocarbon dating (AMS $^{14}$C), which was performed at the University of Erlangen (Germany) and Poznan Radiocarbon Laboratory (Poland) [49]. The radiocarbon dates were calibrated with the SHCal 13 calibration curve [55], and the age-depth model was constructed using the Bayesian accumulation model ("Bacon") [56]. All dates are presented as calibrated years before present (cal yr BP). Subsamples for testate amoeba analysis used in the present study were collected from the peat layer (0–240 cm) at 10 cm intervals at depths of 0–165 cm and every 1–2 cm at the depths of 165–240 cm to ensure equal temporal resolution. The sampling strategy resulted in a total of 63 samples. The same core was previously analyzed for pollen and spores, macro-charcoal, loss-on-ignition (LOI), total organic carbon, and carbon isotopes ($\delta^{13}C_{org}$) (for more details see Hapsari et al. [49,52]).

### 2.4. Testate Amoeba Analysis

Samples for testate amoeba analysis were prepared following the method based on filtration and sedimentation of water suspensions [57]. Each sample was placed in a conical flask with a volume of 250 mL and filled with water. Then, flasks were intensively shaken for 10 min to release testate amoebae from the substrate. After that, the contents of the flasks were carefully sieved through a 500 μm mesh into a beaker (800 mL) and left to settle for 4 h. The supernatant was carefully discarded, and the sediment was transferred to a measuring cylinder (100 mL) and left to settle again for 12 h. Then, the supernatant was carefully discarded to concentrate the sediment to a volume of 10 mL. This concentrate suspension was transferred to a storage container, fixed with neutralized formaldehyde, and used for analysis. Several drops (20 μL) of the concentrate were mounted with glycerol and investigated under a light microscope (Motic BA310T, Xiamen，China) at a magnification of 200–400×. In total, 30 slides (600 μL) were analyzed for each sample to ensure comparability of testate amoeba counts across samples. All encountered testate amoeba shells were identified and counted. Shell length and shell width were measured for each shell to assign it to a particular size class. If the size range corresponded to the original description, we assigned this individual to the type species; otherwise, we assigned individuals into morphotypes of known morphospecies and named them "morph". For all species (morphospecies) of testate amoebae observed in the deposits, a database of functional traits was developed following Krashevska et al. [20] (Table S1). Briefly, the database included two quantitative (shell length and width) and four qualitative (shell shape, shell compression, aperture shape, and aperture invagination) functional traits as the most relevant for the reconstruction of the surface wetness in peatlands [20].

### 2.5. Data Analyses

The data were analyzed and visualized using the R language environment version 4.1.3 [58]. Based on the function trait database, community weighted means (CWMs) were calculated for each sample in the package 'FD' [59]. The CWM values obtained reflect the functional composition, expressed as the mean trait value of all species present in the community weighted by their relative abundance for quantitative traits (shell length and shell width), or the relative abundance of all taxa with the respective trait for qualitative traits [59]. Stratigraphic diagrams of the morphospecies and functional composition of testate amoeba communities were plotted using the package 'analogue' [60]. Quantitative reconstructions of the surface wetness of the peatland were performed as water table depths

(WTDs) using the morphospecies (Weighted Averaging, Inverse) and functional trait-based transfer functions (Multiple Regression) developed by Krashevska et al. [20] in the 'rioja' package [61]. The functional trait-based transfer function was built on five shell characteristics: aperture shape (oval/circular), aperture invagination (slightly invaginated), shell shape (oviform/elongate), shell compression (sub-spherical), and shell width [20]. The reliability of the morphospecies-based reconstructions was assessed by calculating the abundance sum of sub-fossil taxa missing in the training set (with a cut-off value of 15% for good representativeness). The greater the abundance sum of species missing from the training set, the less reliable the reconstructed WTD values. For the functional trait-based transfer function, we tested the differences in the CWM values between the sub-fossil community and the training set using two-sample Student's *t*-tests.

## 3. Results

### 3.1. Overall Characteristics of Testate Amoeba Communities

The analysis of the samples revealed 48 testate amoeba taxa belonging to 20 genera (Figure S1, Table S2). In total, 3595 individuals of testate amoebae were counted and identified. The number of counted shells per sample varied from 3 to 212 and was consistently lower than 50 at depths below 191 cm, which might indicate low initial abundances of testate amoebae due to unfavorable conditions for their development or poor preservation of shells in the sediment (Table S2, Figure 2). The most abundant taxa (with a relative abundance to the total counts greater than 3%) were *Hyalosphenia subflava* morph 2 (22.2%), *H. subflava* morph 9 (16.7%), *H. subflava* morph 12 (15.4%), *H. subflava* morph 14 (5.8%), *H. subflava* morph 16 (4.4%), and *Pyxidicula operculata* morph 1 (3.6%). The most frequently encountered taxa (i.e., observed in more than half of the studied samples, expressed as a percentage of the total number of samples) were *H. subflava* morph 2 (100%), *H. subflava* morph 9 (68.3%), *Cyclopyxis eurystoma* morph 1 (65.1%), *Plagiopyxis callida* (57.1%), *P. operculata* morph 1 (55.6%), *Trigonopyxis arcula* morph 1 (55.6%), and *H. subflava* morph 12 (54.0%). Seven species were encountered in two or fewer samples (with the maximal relative abundance per sample less than 0.1%). The number of taxa per sample varied from 1 to 27 with a mean value of 10.1 (SD = 7.58; n = 63). Overall, despite poor preservation of testate amoeba shells at the bottom layers of the peat deposits, the upper layers of the deposits contained abundant testate amoeba communities dominated by previously undescribed morphospecies.

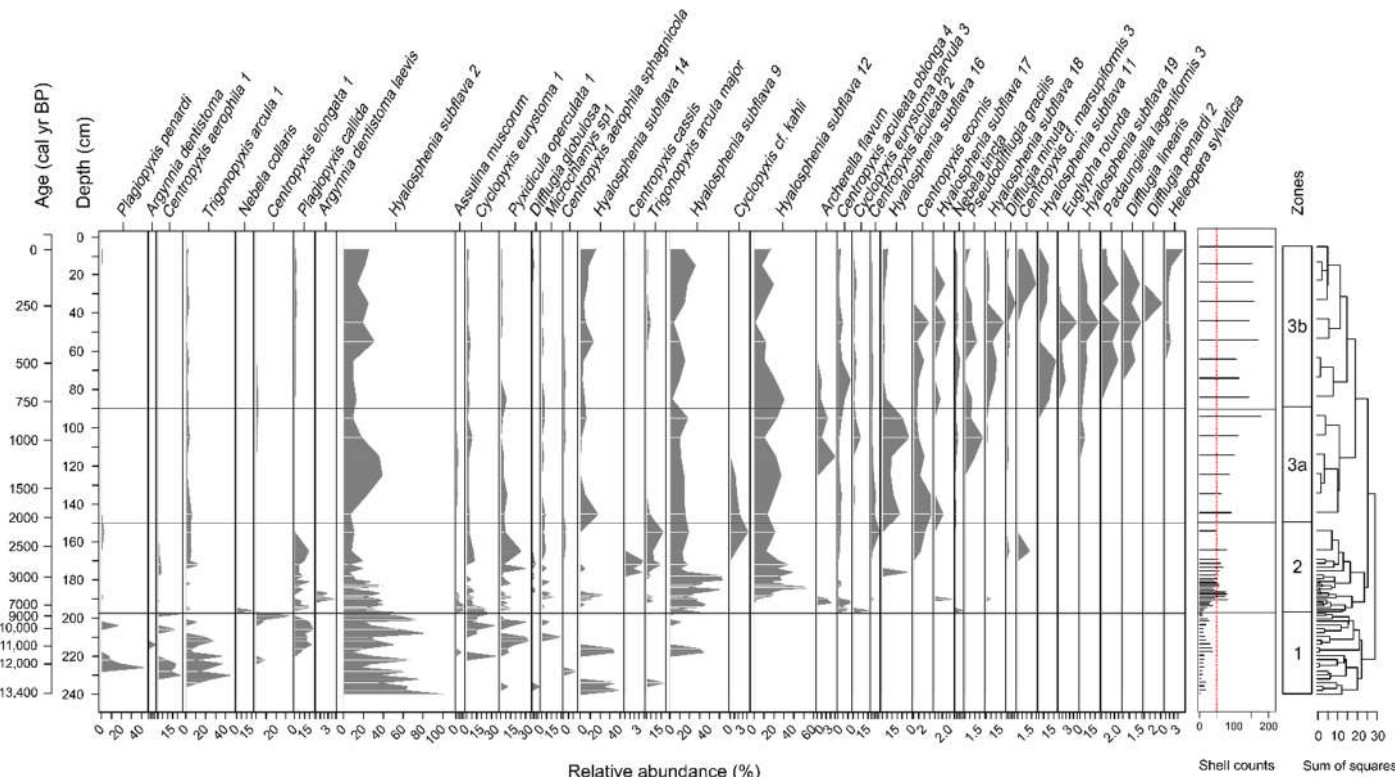

**Figure 2.** Testate amoeba morphospecies (the number after the morphospecies name stands for the morph type) diagram from the Sungai Buluh peatland (Central Sumatra, Indonesia). The zonation is based on a constrained incremental sum of squares. The taxa are ordered by the weighted average of the depth/age axis. For clarity, only taxa observed in five or more samples or with a relative abundance of ≥3% per sample are shown.

### 3.2. Morphospecies-Based Zonation of Peat Deposits and Reconstruction of Water Table Depth

The results of the constrained cluster analysis based on the morphospecies composition of testate amoeba communities indicate that three main zones can be distinguished in the peat deposits: Zone 1: 240–198 cm, 13,400–8000 cal yr BP; Zone 2: 198–150 cm, 8000–2000 cal yr BP; Zone 3: 150–0 cm, 2000 cal yr BP–present (Figure 2). The top zone could be subdivided into two subzones: Zone 3a: 150–90 cm, 2000–750 cal yr BP and Zone 3b: 90–0 cm, 750 cal yr BP–present. All zones were characterized by a high abundance of *Hyalosphenia subflava* morph 2 with the relative abundance varying between 20 and 39% (the highest value in the bottom zone).

Zone 1 (240–198 cm, 13,400–8000 cal yr BP) was characterized by overall low abundances of testate amoebae, which were mostly represented by *Hyalosphenia subflava* morph 2 (39%, here and further to the total counts of testate amoebae in this zone) *H. subflava* morph 14 (11%), *T. arcula* morph 1 (11%), *P. operculata* morph 1 (8%), *H. subflava* morph 9 (7%), *Plagiopyxis callida* (7%), *C. eurystoma* morph 1 (4%), *Centropyxis aerophila* morph 1 (3%), *Plagiopyxis penardi* (3%), and *Centropyxis elongata* morph 1 (2%). Most of these taxa are typical for terrestrial environments (soils and mosses) and/or temporal aquatic habitats. The results of the morphospecies-based quantitative reconstruction indicate high variation in surface wetness (Figure 3).

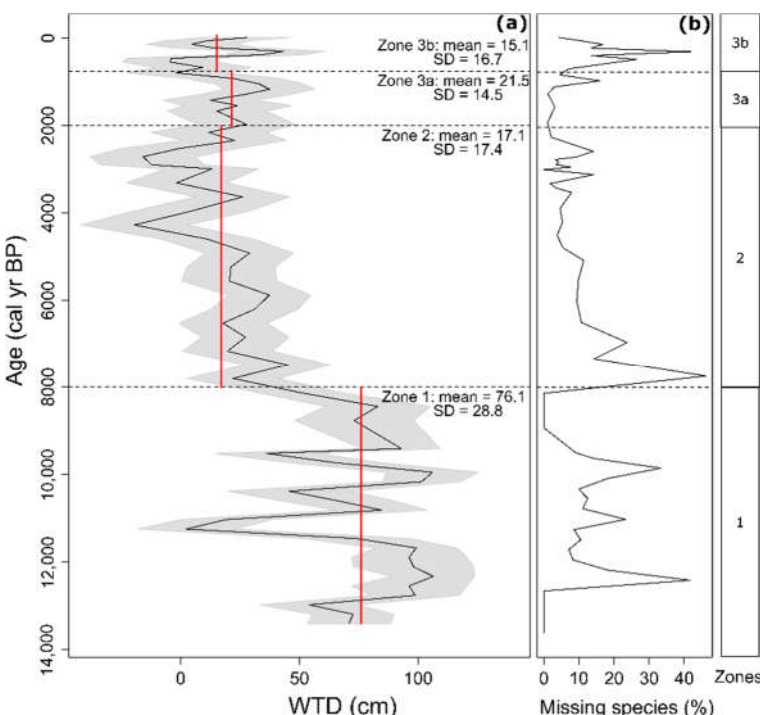

**Figure 3.** Reconstruction of water table depth (WTD, cm) based on morphospecies data (**a**) and quality of the reconstruction (**b**) estimated as abundance sum (%) of species missing from the training set (the greater abundance sums of species missing from the training set, the less reliable the reconstructed WTD values; generally, a cut-off value of 15% for good representativeness can be applied). The zonation is based on the constrained cluster analysis of the morphospecies composition of testate amoeba communities (see Figure 2).

Zone 2 (198–150 cm, 8000–2000 cal yr BP) differed from the previous one by a more consistent presence and greater abundances of *H. subflava* morph 9 (25%) and the appearance of a new morph of *H. subflava* morph 12 (19%). The other typical species were *H. subflava* morph 2 (21%), *P. operculata* morph 1 (6%), *Plagiopyxis callida* (5%), *C. eurystoma* morph 1 (4%), *Trigonopyxis arcula major* (4%), and *H. subflava* morph 14 (3%). Overall, the increased abundances of large morphotypes of *H. subflava* morphs 9 and morphs 12 might be interpreted as increased biotope wetness. Similar trends are shown by the morphospecies-based quantitative reconstruction (Figure 3). However, by the end of the zone (around 2500 cal yr BP), the abundance of hydrophilic taxa decreased in favor of xerophilic and soil-dwelling taxa *Trigonopyxis arcula* morph 1, *Trigonopyxis arcula major*, and *Cyclopyxis* cf. *kahli*, which was associated with the decreasing surface wetness (i.e., greater WTD values).

Zone 3 (150–0 cm, 2000 cal yr BP–the present) was characterized by greater abundance and diversity of testate amoebae community but was still dominated by morphs *H. subflava* morph 2 (20%) of the total counts in the zone, morph 12 (16%), morph 9 (13%), morph 14 (7%), morph 16 (7%), morph 11 (5%), morph 18 (4%), and morph 19 (4%). The other species (with a relative abundance greater or equal 1%) were *Centropyxis aculeata oblonga* morph 4 (3%), *C. eurystoma* morph 1 (1%), *Cyclopyxis eurystoma parvula* morph 3 (2%), *Difflugia minuta* (2%), *Nebela tincta* (2%), *P. operculata* morph 1 (2%), *T. arcula* morph 1 (2%), *Archerella flavum* (1%), *Centropyxis aculeata* morph 2 (1%), *Centropyxis ecornis* (1%), *Euglypha rotunda* (1%), *Heleopera sylvatica* (1%), *Padaungiella lageniformis* morph 3 (1%), *Plagiopyxis callida* (1%), *Pseudodifflugia gracilis* (1%), and *T. arcula* major (1%). The zone could be subdivided into two subzones at the depth of 90 cm (750 cal yr BP). Upper subzone 3b (90–0 cm, 750 cal yr BP–present) was characterized by the appearance of new taxa, lower relative abundances of *H. subflava* morph 16 (−14%) and morph 12 (−5%), and greater relative abundances of morph 11 (+7%), morph 18 (+5%) and morph 19 (+5%) as compared

to subzone 3a. The quantitative reconstruction indicates a slight tendency to have a decreased surface wetness in subzone 3a and a greater variation in subzone 3b (Figure 3).

### 3.3. Functional Trait-Based Zonation of Peat Deposits and Reconstruction of Water Table Depth

The functional composition of the sub-fossil testate amoeba community (Figure 4) varied in accordance with the bio-stratigraphic zones defined based on morphospecies composition with the most considerable changes at the border between Zones 2 and 3. Overall, community-weighted means (CWMs) for most of the functional traits were more variable in Zones 1 and 2, as compared to Zone 3. Shell length increased from Zone 1 (mean CWM 63.0 ± 8.6 μm SD) to Zone 3b (79.4 ± 8.1 μm), whereas shell width remained relatively constant through the entire deposits (49.6 ± 8.6 μm). Circular shells were more abundant in Zone 1 and Zone 2 (34.4 ± 24.3 and 25.7 ± 16.5%, respectively), as compared to Zone 3 (17.9 ± 5.7%). On the contrary, the proportion of oviform shells increased from 62.4 ± 23.7% in Zone 1 to 80.6 ± 5.0% in Zone 3b. Oviform-elongate shells did not show any clear patterns, with just a single peak (up to 33%) at the border between Zone 1 and Zone 2, whereas flask-shaped shells were typical for Zone 3b (less than 2%). The relative abundance of shells with hemispheric and sub-spherical compression was the greatest in Zone 1 (16.0± 16.9 and 25.8 ± 23.6%, respectively) and decreased to the top (6.2 ± 2.8 and 2.1 ± 2.4%, respectively). Compressed (82.2 ± 5.3%) and spherical shells (21 ± 2.3%) were more typical for Zone 3. Very compressed shells appeared at the beginning of Zone 3b (less than 2%) for a short period (760–520 cal yr BP). Shells with irregular and slit-like apertures dominated in Zone 1 (13.0 ± 14.3 and 9.3 ± 11.0%, respectively), whereas shells with oval apertures were consistently abundant in Zone 3 (80.3 ± 5.3%) in comparison to the lower and more variable values in Zones 1 and 2 (62.4 ± 23.7% and 74 ± 16.3%, respectively). The proportion of shells with non-invaginated apertures increased from 63.7 ± 24.2% in Zone 1 to 85.1 ± 4.9% in Zone 3b. Both Zone 1 and 2 were characterized by greater proportions of the shells with slightly- and strongly invaginated apertures (5.5 ± 8.1 and 7.1 ± 8.5%, respectively), which were less abundant in Zone 3 (3.5 ± 2.7% and 0.6 ± 0.6%, respectively).

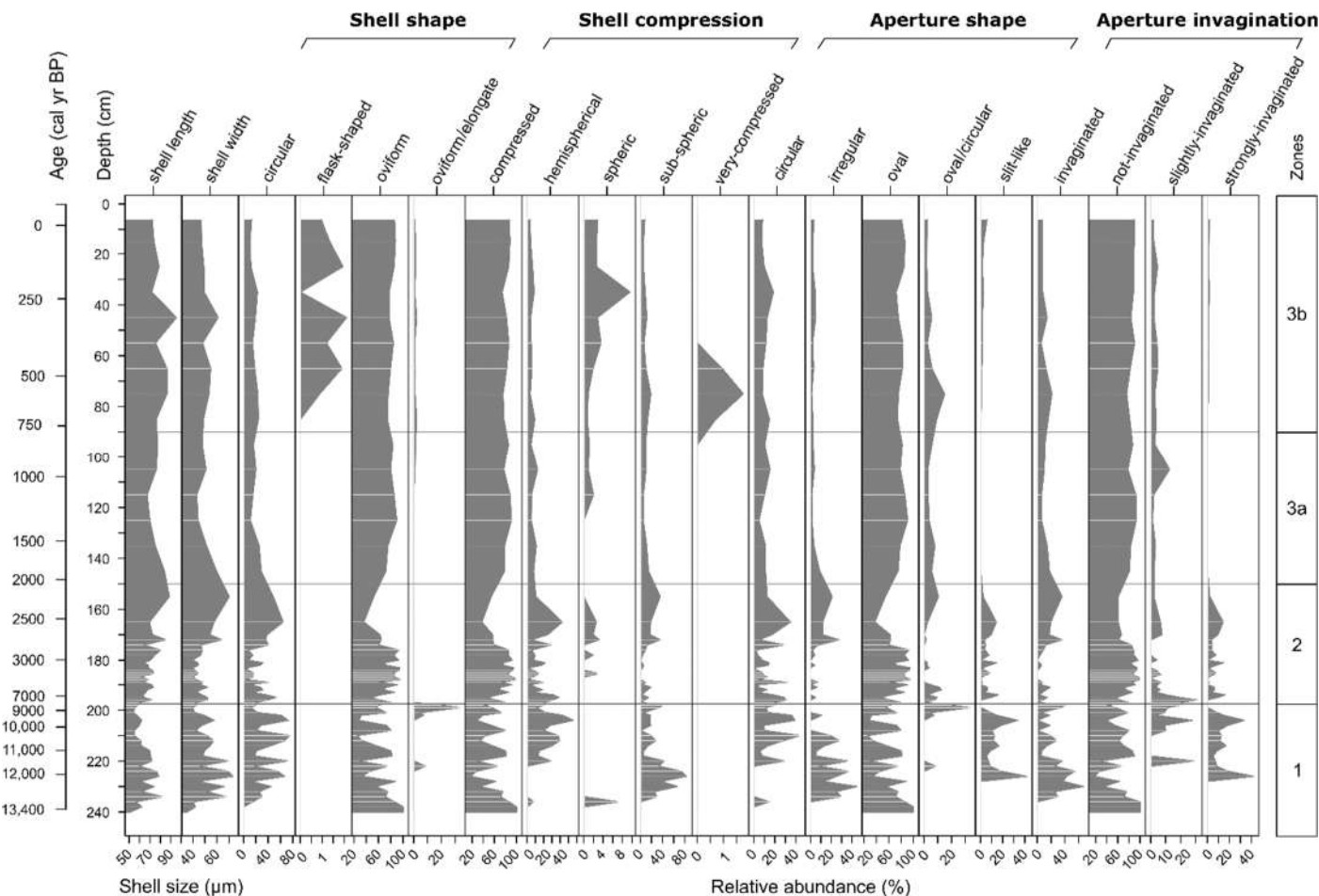

**Figure 4.** Community-weighted means for functional traits of testate amoeba communities from the peat deposits of the Sungai Buluh peatland (Central Sumatra, Indonesia). The zonation is based on the constrained cluster analysis of the morphospecies abundance data (see Figure 2).

The quantitative reconstruction with the functional trait-based transfer function, which included five traits (aperture shape (oval/circular), aperture invagination (slightly invaginated), shell shape (oviform/elongate), shell compression (sub-spherical), and shell width) showed poor performance and predicted unrealistic values (Figure 5). Correlation between the morphospecies- and functional-trait-based reconstructions was weak and only marginally significant (Spearman's correlation rho = 0.22, *p* = 0.08). This relationship was even weaker and non-significant when calculated only for the samples where more than 50 shells were counted (Spearman's correlation rho = –0.06, *p* = 0.73). However, the functional trait-based reconstruction also indicates a clear trend of increased surface wetness (i.e., lower WTD) over the period of peatland development. Besides the low shell counts in the bottom peat layers, the poor performance of the functional trait transfer function might be explained by the low overlap between ranges of CWM for the modern and sub-fossil communities (Figure 6). Sub-fossil communities have greater shell width (49.6 ± 8.57 μm) as compared to modern ones (38.1 ± 3.54 μm). Furthermore, the proportion of the shells with oval/circular aperture (3.6 ± 5.73%) and sub-spherical shell compression (15.0 ± 17.5%) was greater in the sub-fossil communities in comparison to the modern ones (0.8 ± 1.14%, and 3.1 ± 12.6%, respectively), whereas oviform/elongate shells were more abundant in the modern dataset (3.6 ± 4.88%) as compared to the sub-fossil communities (1.3 ± 5.0%).

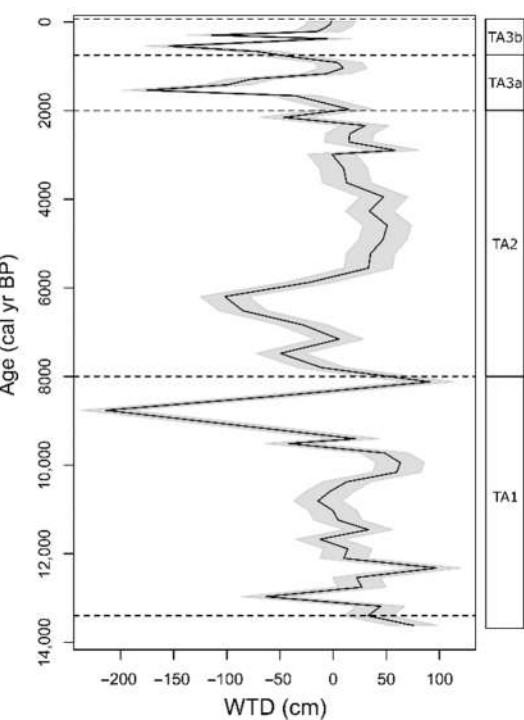

**Figure 5.** Reconstruction of water table depths (WTD, cm) based on the functional trait composition of testate amoeba communities using the multiple regression transfer function. The zonation is based on the constrained cluster analysis of the morphospecies composition of testate amoeba communities (see Figure 2).

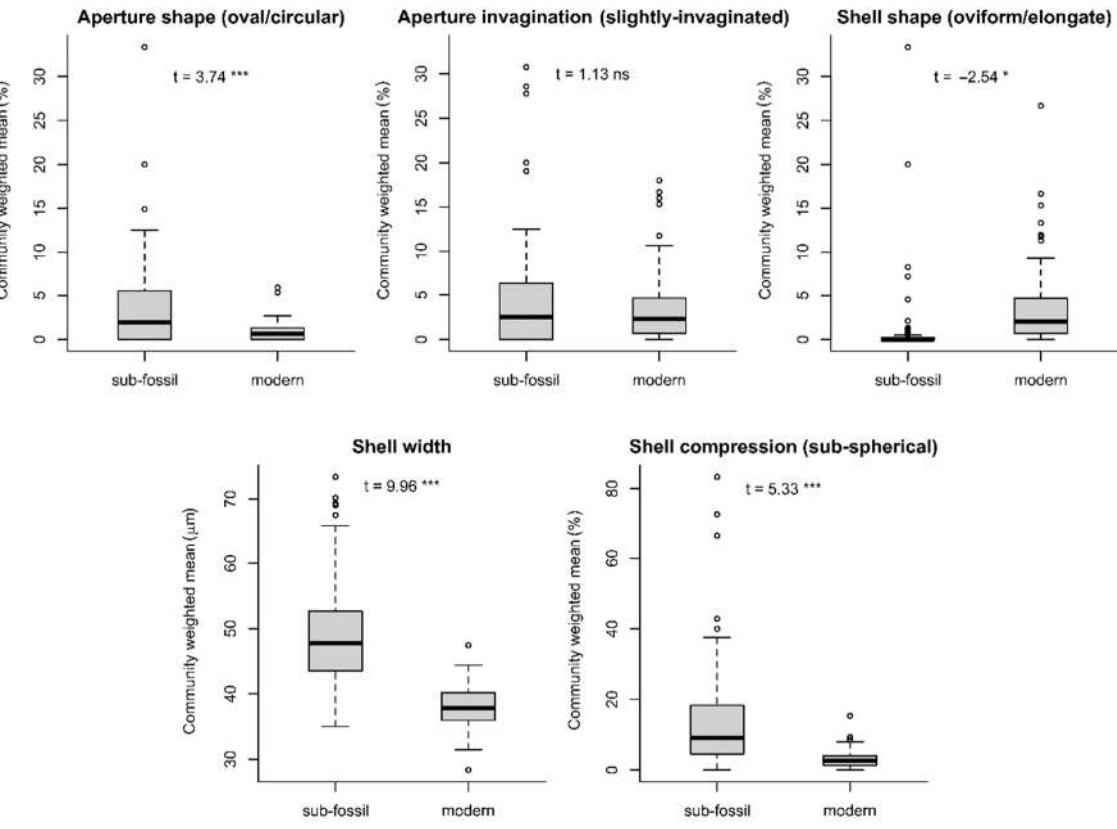

**Figure 6.** Community-weighted means (CWMs) for the modern (training set) and sub-fossil community of testate amoebae in the Sungai Buluh peatland (Central Sumatra, Indonesia). The results of a two-sample *t*-test are added to the plots to show significant differences (ns—not significant, *—$p < 0.05$, ***—$p < 0.001$).

## 4. Discussion

The results of the present study allow us to describe the species composition of sub-fossil testate amoebae and to reconstruct the hydrological regime of a tropical peatland during the Holocene. Despite the poor preservation of testate amoeba shells in bottom peat layers, we could define three main stages in the development of the peatland, which correspond to the previously collected paleoecological information based on other proxies (pollen, organic matter content, [13]C isotopes, etc. [49]). We also discuss the applicability of morphospecies- and functional-trait-based approaches, showing that the latter has weak sides as well and cannot be considered as a panacea and a complete replacement for the former.

### 4.1. Initial Stages of Peatland Development and Topogeneous Stage (13,400–8000 cal yr BP)

The initial stage (Zone 1: 240–198 cm, 13,400–8000 cal yr BP) of the peatland development was characterized by the presence of taxa typical for terrestrial environments (soils and mosses) and/or temporal aquatic habitats. It has been previously shown that *Pyxidicula operculata* prefers aquatic habitats [62]. *Pyxidicula* has also been documented at the early development stages in some peatlands of temperate zones [40,63] or habitats with high groundwater [36] as well as in tropical peatlands [11]. The species *Plagiopyxis callida* and *Plagiopyxis penardi* are typically soil dwellers [64,65], whereas *Centropyxis aerophila* and *Centropyxis elongata* are common in terrestrial habitats with mosses and humus-rich soils [66]. Furthermore, *C. aerophila* was also documented in freshwater habitats [67]. The presence of small eurybiotic (i.e., species with wide ecological preferences) *Cyclopyxis eurystoma* in peat deposits is considered as an indication of unstable environmental conditions [11]. However, Krashevska et al. [20] estimated the ecological optimum of *C. eurystoma* to WTD to be 56–62 cm, which was comparable to the xerophilic *Trigonopyxis arcula* morph 1 (62 cm) that also dominated in the zone. All these taxa were also documented in tropical peatlands of central Sumatra [11,20]. These results are in line with previous studies [49], which showed that the peat deposits are underlined by the basal clay materials of terrestrial origins (as shown by $\delta^{13}C_{org}$ value of ca. 28‰), in contrast to most of the other peatlands in Indonesian coastal areas, which are developed over mangrove or marine sediments [49]. This implies that the organic materials were accumulated on a floodplain depression that was filled by a river impediment and/or rising water table due to the rapid sea level rise and was characterized by the dominance of mixed-riverine forests (based on pollen data), low organic matter content (LOI < 75%), and slow peat accumulation rate (0.1 mm yr$^{-1}$) [49]. Overall, the dominance of taxa typical for soils and aquatic biotopes from 13,400 8000 cal yr BP supports the previous findings that the initial stage of the peatland development can be considered topogeneous.

Our results of species-based reconstruction indicate dry but highly variable surface wetness conditions during the initial stage of peatland formation. However, the relatively high reconstructed values of WTD should be interpreted with caution due to low shell counts in the zone and few aquatic biotopes included in the training set [20]. The hydrological preferences of two dominant morphs of *Hyalosphenia subflava* (morph 2 with an average shell length of 51 μm and morph 14 with an average shell length of 79 μm) to WTD were previously estimated as 73.6 cm and 49.1 cm, respectively [20]. The former was consistently present in Zone 1, whereas the latter appeared just for short periods that might indicate wet conditions [20,68]. The functional trait composition in Zone 1 was characterized by a predominance of short shells with irregular and slit-like apertures that were slightly or strongly invaginated, which also indicates variable surface wetness. Previous studies demonstrated that testate amoeba with small shells dominate in communities typical for dry peatland biotopes or biotopes with unstable surface wetness in North America and Europe [30,36,39]. However, these relationships seem to be non-linear, and some large species (such as *Bullinularia indica* and *Trigonopyxis arcula*) have other adaptations (e.g.,

aperture form) to dry habitats, but they never seem to dominate communities to shift average shell size to greater values. Greater proportions of taxa with slit-like apertures (compressed acrostomic type) were also reported for dry biotopes [69]. Similar preferences are typical for shells with invaginated apertures because this is generally interpreted as an adaptation to low substrate moisture [36]. Overall, both morphospecies and function traits indicate fluctuating WTDs at this stage that might be related to unstable water regimes in floodplains [70] that are influenced by several factors such as river development stage, distance from the river, rainfall, and flood events [70–72].

### 4.2. Transitional Stage (8000–2000 cal yr BP)

Morphospecies composition of testate amoeba communities in Zone 2 (198–150 cm, 8000–2000 cal yr BP) indicate greater and more stable surface wetness of the peatland. Most of the species from Zone 1 remained in the community but reduced their abundances in favor of larger morphotypes of *Hyalosphenia subflava* (morph 9 and 12), which prefer wetter biotopes with average WTD values of 35.1 and 42.6 cm [20]. In terms of functional trait composition of testate amoebae, Zone 2 can be considered transitional. This finding corroborates the results of the previous study (Hapsari et al., 2017), which related the deposition of clayey peat (i.e., peat with an admixture of clay materials, LOI > 75%) with the frequent river floods due to ineffective drainage as a result of sea level rise [73] and high precipitation [74]. By the end of the zone, around 2500 cal yr BP, the abundance of large morphotypes of *Hyalosphenia subflava* decreased, which was associated with a slight tendency to show a drier peatland surface. At the same time, mixed-riverine vegetation was gradually replaced by the peat swamp forest (similar to that present at the site nowadays) [49]. These changes were related to reduced flooding frequency due to the increased isolation of the peatland from the river flood impacts [75], sea level regression [76], and reduced precipitation [77]. In addition, these processes were associated with increased acidity [49] of the environment, which eventually resulted in structural shifts in the testate amoeba community.

### 4.3. Ombrogenous Stage (2000–750 cal yr BP)

The next stage of the peatland development (Zone 3a: 150–90 cm, 2000–750 cal yr BP) was associated with the greater presence of taxa that are generally considered moss-dwelling (e.g., *Nebela tincta*, *Trigonopyxis arcula*, *Euglypha rotunda*, and *Padaungiella lageniformis*) but in this environmental context can be interpreted as indicators of acidic conditions. The results of the surface wetness reconstruction support the previously reported stabilization of the hydrological regime [49], which was inferred based on maximum coverage of peat swamp forests (by 1200 cal yr BP) and increased peat accumulation rates. Low variation in surface wetness reduces the oxygen supply, leading to slower rates of aerobic decay of organic matter [78]. Furthermore, less river flooding could also increase cambial growth, which results in greater biomass production [79]. Altogether, this subsequently led to peat thickening and gave rise to ombrogenous peat. In these conditions, the proportion of circular shells decreased in favor of oviform shells; hemispheric and sub-spheric compressed shells were replaced by very compressed shells, while oval non-invaginated apertures became the most dominant. Oviform shells are generally associated with epiphytic habitats [36], which also favor testate amoeba communities with greater proportions of compressed shells. Shells with non-invaginated apertures are common in wet biotopes [36].

### 4.4. Ombrogenous Stage and Anthropogenic Impacts (750 cal yr BP–Present Days)

A slight shift in testate amoeba composition towards a greater diversity and variation in the relative abundance of the dominant morphospecies took place around 750 cal yr BP. In terms of the functional trait composition, testate amoeba communities were characterized by a greater abundance of flask-shaped shells (mostly due to the presence of

*Padaungiella lageniformis*) throughout the entire zone, very compressed shells at the beginning, and the shells with spherical compression at the end of the zone. Although the functional role of flask-shaped shells remains unclear, shell compression is generally interpreted in terms of adaptation to survive in thin water films during water-deficient stages so that amoebae can stay active longer when the substrate moisture content decreases [36]. This is corroborated by the results of the previous reconstruction at the site showing that the environmental changes at this stage might be attributed to forest opening, as shown by pollen data [49,52], associated with the human impact of an ancient kingdom, the Malayu Empire (1100–600 cal yr BP). The people of the Malayu Empire actively used wood and plant materials for building purposes [50]. Pollen data [49,52] indicate a forest recovery after 600 cal yr BP when the Malayu Empire collapsed [51]. These activities in the area could change the water regime of the peatland by reducing evapotranspiration and increasing surface erosion in adjacent areas that could create niches for an introduction of new testate amoeba taxa. Previous studies have demonstrated that the functional composition of testate amoeba communities might change in response to human impacts by reducing the relative abundance of mixotrophs [36] or by increasing the dominance of small taxa [30]. However, considering the ranges of potential environmental changes associated with human activities, all these functional traits might be important for tracing them.

### 4.5. Morphospecies- vs. Functional-Trait-Based Reconstructions

The application of testate amoebae to paleoecological reconstructions based on peatland deposits in the Southern Hemisphere in general and tropical peatlands in particular is still limited by the spatial extent of the existing research [20,24,29]. The functional-trait-based approach was generally considered as an opportunity to overcome the limitations of the morphological-based approach; however, our results indicate that the application of the former might be also limited. Both approaches showed a good performance during the cross-validation procedures according to Krashevska et al. [20]. Nonetheless, the functional trait-based quantitative reconstruction of the water table depths in the Sungai Buluh peatland poorly corresponded to the results of the morphospecies-based reconstruction, which turned out to be more realistic. Besides the poor preservation of shells in the bottom layers, the possible explanation of the weak performance of the functional trait-based transfer function might be the multiple linear regression technique, which is generally used for modelling relationships between relatively small number of functional traits and a single environmental variable. On the one hand, this helps to simplify the computation and avoid multicollinearity among explanatory variables (in this case functional traits), but this method seems to perform well only over short gradients with a predominantly linear relationship between traits and environment [80]. Our sub-fossils data demonstrated a greater variation in the functional traits as compared to the training set data, even in the top (recent) peat layers. This might be related to more diverse environmental conditions in the Holocene history of the Sungai Buluh peatland [49], while the training set was limited by the modern state of the peatland only. Moreover, the relationships between the function traits of testate amoebae are not always linear, e.g., large shells of xerophilic *Bullinularia indica* and *Trigonopyxis arcula*. Further, the functional traits are less numerous than taxa, which increases the risk of loss or scarcity of comparable traits for analyses. In this context, the morphospecies-based transfer function performed better because it relies on the unimodal relationships between environment and morphospecies and a greater number of taxa. Overall, functional traits provide useful information for the interpretation of paleoecological records. However, functional-trait-based quantitative reconstructions might require further research on building training sets covering a greater diversity of environmental conditions and the application of more appropriate modelling techniques.

## 5. Conclusions

By applying morphospecies- and functional-trait-based approaches to the reconstruction of the hydrological regime of a tropical peatland during the Holocene, this study demonstrates a high indicator value of testate amoebae in these environmental settings. Three main periods of peatland development can be distinguished based on the morphospecies composition of testate amoebae: 13,400–8000, 8000–2000, and 2000 cal yr BP–present. This zonation corresponds well to the previously described periods. Moreover, testate amoebae responded to human-related deforestation of the peatland around 750 cal yr BP by greater abundances of flask-shaped shells through the entire zone, very compressed shells in the beginning, and shells with spherical compression at the end of the zone. Functional traits provided useful information for the overall qualitative interpretation of the testate amoebae record; however, the quantitative reconstruction based on the functional-trait transfer function was less reliable than the morphospecies-based one. This is partly due to our poor and still limited knowledge of the relationships between environmental characteristics and functional traits, especially in tropical environments. Thus, the functional-trait-based approach cannot completely replace the morphological one for quantitative environmental reconstructions, at least at the present stage. Despite the advantages of the former, it still can be subjected to the risks of poor shell preservation and coverage in the training set and modeling issues. Future studies on the functional and morphospecies composition of testate amoebae in a wider range of tropical peatlands could considerably improve the quality of hydrological reconstructions.

**Supplementary Materials:** The following supporting information can be downloaded at: www.mdpi.com/article/10.3390/d14121058/s1, Table S1: Functional traits of testate amoeba species in the sub-fossil communities from peat deposits of Sungai Buluh peatland (Central Sumatra, Indonesia); Table S2: Counts of testate amoebae in peat deposits of Sungai Buluh peatland (Central Sumatra, Indonesia); Figure S1: The main morphospecies of testate amoebae observed in the peat deposits of Sungai Buluh peatland (Central Sumatra, Indonesia).

**Author Contributions:** Conceptualization, V.K., A.N.T.; provided field logistic support, sampling and samples preparation, K.A.H., S.B., S.S.; software, A.N.T.; testate amoebae analysis and data supervision, E.A.M., Y.A.M.; contributed reagents, material, and analysis tools V.K, Y.A.M., H.B., S.B., S.S.; data curation V.K., A.N.T., Y.A.M.; writing—original draft preparation, V.K., A.N.T.; writing—review and editing, S.B., K.A.H.; visualization, V.K., A.N.T.; funding acquisition, Y.A.M., H.B. All authors contributed to writing and editing the manuscript and gave final approval for publication. All authors have read and agreed to the published version of the manuscript.

**Funding:** This research was funded by the Deutsche Forschungsgemeinschaft (DFG, German Research Foundation), grant number 192626868, in the framework of the collaborative German-Indonesian research project CRC990–EFForTS (V.K., K.A.H., S.B., S.S. and H.B.), and by the Russian Science Foundation, grant number 19-14-00102 (A.N.T. and Y.A.M., testate amoeba analysis and manuscript preparation).

**Institutional Review Board Statement:** This study was conducted using the research permit for subproject A01 (RISTEK; 23/EXT/SIP/FRP/E5/Dit.KI/VI/2017) from the Ministry of Research and Technology of Indonesia and sample export permit (B-1127/IPH.1/KS.02.04/111/2019) based on the recommendation of Indonesian Institute of Sciences (LIPI).

**Data Availability Statement:** The data supporting the reported results can be found in the Supplementary Materials to this paper.

**Acknowledgments:** We gratefully acknowledge the logistic support by the EFForTS coordination team and the Indonesian partner universities in Bogor and Jambi, Institut Pertanian Bogor (IPB) and University of Jambi (UNJA), the Ministry of Education in Jakarta (DIKTI), and the Indonesian Institute of Sciences (LIPI).

**Conflicts of Interest:** The authors declare no conflicts of interest. The funders had no role in the design of the study; in the collection, analyses, or interpretation of data; in the writing of the manuscript; or in the decision to publish the results.

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
