# Peer review of "Species- and Trait-Based Reconstructions of the Hydrological Regime in a Tropical Peatland (Central Sumatra, Indonesia) during the Holocene Using Testate Amoebae"

_diversity, doi:10.3390/d14121058_

Round 1
Reviewer 1 Report
Main comments: The authors have provided a well-written manuscript that provides some new insights into the palaeoecology of tropical testate amoebae from Indonesia. My major concern is that the testate amoebae counts are too low to derive meaning inferences from both the species and functional trait-based reconstruction models. Only 12 samples are above 100 testate amoebae sub-fossils in the top portion of the core (-15 to 1044 cal. yr BP), with the remaining samples well below the minimum test count needed to derive meaningful statistical inferences. I understand that this is due to low preservation of testate amoebae, and I suggest that the authors focus less on the model in the core that is older than 1044 cal. yr BP. Instead, a more qualitative description of relative abundance through time should be provided instead, taking into account the preservation issues. However, the model could be used in the upper portion of the core, where the inferences are more statistically sound. The focus on the model, in my opinion, detracts from the study due to the low performance in many samples. While there is good work in this paper, I feel that the manuscript needs to be re-written.
Moderate comments:
lines 47-64. There is a focus on carbon storage in tropical systems. I understand that this is setting the scene for the paper, but I think it can be shortened to get to the point regarding hydrological regimes and how this effects tropical peatland carbon storage.
Lines 65-67. I completely agree that there has been far less focus placed on tropical peatlands; however, it would be worth scrutinising this further and talking about the work that has been done briefly or even just mentioning some papers. I’ve included a few below.
Fournier, B., Coffey, E.E., van der Knaap, W.O., Fernández, L.D., Bobrov, A. and Mitchell, E.A., 2016. A legacy of human‐induced ecosystem changes: spatial processes drive the taxonomic and functional diversities of testate amoebae in Sphagnum peatlands of the Galápagos. Journal of Biogeography, 43(3), pp.533-543.
Barrett, K.D., Sanford, P. and Hotchkiss, S.C., 2021. The ecology of testate amoebae and Cladocera in Hawaiian montane peatlands and development of a hydrological transfer function. Journal of Paleolimnology, 66(2), pp.83-101.
Swindles, G.T., Lamentowicz, M., Reczuga, M. and Galloway, J.M., 2016. Palaeoecology of testate amoebae in a tropical peatland. European Journal of Protistology, 55, pp.181-189.
Liu, B., Booth, R.K., Escobar, J., Wei, Z., Bird, B.W., Pardo, A., Curtis, J.H. and Ouyang, J., 2019. Ecology and paleoenvironmental application of testate amoebae in peatlands of the high-elevation Colombian páramo. Quaternary Research, 92(1), pp.14-32.
Lines 67-70. I recommend introducing this sentence before going in to talk about the fact that they are poorly investigated. Set the scene for why they are a valuable proxy and then mention that, however, only a small number of studies has explored their potential in tropical settings.
Lines 161-163: A 200 x magnification seems very low to be making identifications on smaller specimens. Is this correct? Was a more powerful objective not used? If not, please mention this.
Line 202-208. It’s not clear what the percentage in brackets is referring to? H. subflava morph 2 (100%) – does this mean it was observed in all of the sub-samples from the core. If so, be more specific.
Line 199. The total number of counted shells per samples varied from 3 to 212. Can you give more information on the number of samples that yielded a count <100, as this will have a major potential for the information you can yield from your dataset? A sample with only three specimens is not going to derive meaningful information. This should also be noted in your figures with samples less than 100 specimens identified.
Lines 238-241 and Figure 3. The boundary marking the transition from TA1 to TA2 is driven by a shift from 10 (8124 cal. yr BP; 197 cm dept) to 13 (7803 cal. yr BP; 196 cm dept) morphotypes, with the former sub-sample composed of only three morphotypes and the latter with five morphotypes. Thus, I am sceptical that this is a true transition boundary. I reiterate my previous major concern, that the model cannot adequately reconstruct testate amoebae inference with such a low count of tests.
Lines 269-293: The functional trait approach may not have yielded adequate results due to low count numbers. I think this section should be described using a qualitative rather than quantitative approach.
Figure 4. Zone 1 and Zone 2 show erratic trend. This may be due to low test count.
Discussion section: The section must be re-written to reflect the major challenge with test preservation issues. This should be noted at the start of the section. Reconstructed environmental conditions should be discussed cautiously. The authors should just focus on the testate amoebae autecology, which they have done but be critical of their inference models.
Comments on Figures:
Figure 1 Can you provide a better map with contours.
Figure 2. Bottom x-axis (relative abundance %) not legible. Morphotypes and morphospecies names are hard to read. Can you make them vertical and larger? I don’t like the shaded lines; can you make these bars instead. Also, the samples with low testate amoebae counts (<50 sub-fossils) should be highlighted.
A figure with photomicrographs of the specimens should also be included in the manuscript.
Author Response
Dear Editors and Reviewers,
Thank you for the critical review of our manuscript diversity-1985853 entitled " Species and Trait-Based Reconstructions of the Hydrological Regime in a Tropical Peatland (Central Sumatra, Indonesia) during the Holocene Using Testate Amoebae". We have carefully evaluated the comments and suggestions and have made necessary changes to the manuscript, which are detailed below in our point-by-point replies.
Reviewer 1
Comments and Suggestions for Authors
Main comments: The authors have provided a well-written manuscript that provides some new insights into the palaeoecology of tropical testate amoebae from Indonesia.
Re: We appreciated your feedback, all the comments and the time invested for reviewing the manuscript
My major concern is that the testate amoebae counts are too low to derive meaning inferences from both the species and functional trait-based reconstruction models. Only 12 samples are above 100 testate amoebae sub-fossils in the top portion of the core (-15 to 1044 cal. yr BP), with the remaining samples well below the minimum test count needed to derive meaningful statistical inferences. I understand that this is due to low preservation of testate amoebae, and I suggest that the authors focus less on the model in the core that is older than 1044 cal. yr BP. Instead, a more qualitative description of relative abundance through time should be provided instead, taking into account the preservation issues. However, the model could be used in the upper portion of the core, where the inferences are more statistically sound. The focus on the model, in my opinion, detracts from the study due to the low performance in many samples. While there is good work in this paper, I feel that the manuscript needs to be re-written.
Re: Indeed, in tropical peatland it difficult to get above 100 shells per sample that maybe due to various reasons including low preservation. We transparently stated this now and it reads: “The number of counted shells per sample varied from 3 to 212 and was consistently lower than 50 at depths below 191 cm that might indicate low initial abundances of testate amoebae due to unfavorable conditions for their development or poor preservation of shells in the sediment (Figure 2; Table S2).“ It has been demonstrated elsewhere that a meaningful as was shown before paleoenvironmental signal may still predominate over random noise and major changes for counts of less than 100 but more than 50 (Payne and Mitchell, 2009). In our case a half of the samples reach this amount. Further, in last publication from the same region, even low amount of testate amoebae shells reflected/predicted similar hydrology as compared to plant community (Biagioni et al 2015. doi:10.1016/j.palaeo.2015.09.048). Additionally, we added to Fig 2 the shell count for each depth to make our date even more transparent.
As suggested by the reviewer, we provided a thorough qualitative description of subfossil testate amoeba morphotypes in the results section and interpret ecological preferences of those taxa in the discussion. So, we believe species-based information for each zone is of great value especially of this understudied region. We did an extra correlation test between morphotype and function trait based reconstructed values of WTD only for the samples with the counts greater than 50 shells. The results of the test supported our previous conclusions about weak correlations between those variables. We added the description of the test in the results “Correlation between the morphotype- and functional trait-based reconstructions was weak and only marginally significant (Spearman's correlation rho = 0.22, p = 0.08). This relationship was even weaker and non-significant when calculated only for the samples where more than 50 shells were counted (Spearman's correlation rho = -0.06, p = 0.73).”
Moderate comments:
lines 47-64. There is a focus on carbon storage in tropical systems. I understand that this is setting the scene for the paper, but I think it can be shortened to get to the point regarding hydrological regimes and how this effects tropical peatland carbon storage.
Re: We changed as suggested. We deleted following sentence: “The estimates of the carbon deposits in these systems were also raised to 152-288 Gt [4].” We add new link from hydrology to carbon at the end as suggested. It reads now “However, despite the fact the hydrological regime is one of the main factors affecting peatlands, which also determines the storage and flux of carbon in these ecosystems, it has been rarely reconstructed in paleoecological studies on tropical peatlands. “
Lines 65-67. I completely agree that there has been far less focus placed on tropical peatlands; however, it would be worth scrutinising this further and talking about the work that has been done briefly or even just mentioning some papers. I’ve included a few below.
“Fournier, B., Coffey, E.E., van der Knaap, W.O., Fernández, L.D., Bobrov, A. and Mitchell, E.A., 2016. A legacy of human‐induced ecosystem changes: spatial processes drive the taxonomic and functional diversities of testate amoebae in Sphagnum peatlands of the Galápagos. Journal of Biogeography, 43(3), pp.533-543.
Barrett, K.D., Sanford, P. and Hotchkiss, S.C., 2021. The ecology of testate amoebae and Cladocera in Hawaiian montane peatlands and development of a hydrological transfer function. Journal of Paleolimnology, 66(2), pp.83-101.
Swindles, G.T., Lamentowicz, M., Reczuga, M. and Galloway, J.M., 2016. Palaeoecology of testate amoebae in a tropical peatland. European Journal of Protistology, 55, pp.181-189.
Liu, B., Booth, R.K., Escobar, J., Wei, Z., Bird, B.W., Pardo, A., Curtis, J.H. and Ouyang, J., 2019. Ecology and paleoenvironmental application of testate amoebae in peatlands of the high-elevation Colombian páramo. Quaternary Research, 92(1), pp.14-32.
Re: We add all suggested references. It reads now:
“Among the proxies for reconstruction of the hydrological regime in mires are peat humification, hydrogen isotopes [13], and testate amoebae have great potential in tropical settings (Barrett et al 2021; Fournier et al 2016; Lui et al 2019; Swindles et al. 2016).”
And here:
“Only a few studies used testate amoeba for paleoecological reconstructions or development of transfer functions in the tropical peatlands, e.g. montane peatlands (Barrett et al 2021; Lui et al 2019), lowland Amazonian peatlands (Swindles et al. 2014; 2016; 2018) and Indonesian peatlands (Biagioni et al 2015; Krashevska et al 2020).
Lines 67-70. I recommend introducing this sentence before going in to talk about the fact that they are poorly investigated. Set the scene for why they are a valuable proxy and then mention that, however, only a small number of studies has explored their potential in tropical settings.
Re: Thank you for the comment. Changed as suggested. See added changes above (comment to the line Lines 65-67)
Lines 161-163: A 200 x magnification seems very low to be making identifications on smaller specimens. Is this correct? Was a more powerful objective not used? If not, please mention this.
Re: Thank you for the comment. We add missing information: “200-400x”.
Line 202-208. It’s not clear what the percentage in brackets is referring to? H. subflava morph 2 (100%) – does this mean it was observed in all of the sub-samples from the core. If so, be more specific.
Re: Yes, with 100 % meaning that it was observed in all samples. We add the explanation: “The most frequently encountered taxa (i.e., observed in more than a half of the studied samples, expressed as percentage to the total number of samples)”
Line 199. The total number of counted shells per samples varied from 3 to 212. Can you give more information on the number of samples that yielded a count <100, as this will have a major potential for the information you can yield from your dataset? A sample with only three specimens is not going to derive meaningful information. This should also be noted in your figures with samples less than 100 specimens identified.
Re: We added the reference to the Table S2, and the shell counts to the Figure 2.
Lines 238-241 and Figure 3. The boundary marking the transition from TA1 to TA2 is driven by a shift from 10 (8124 cal. yr BP; 197 cm dept) to 13 (7803 cal. yr BP; 196 cm dept) morphotypes, with the former sub-sample composed of only three morphotypes and the latter with five morphotypes. Thus, I am sceptical that this is a true transition boundary. I reiterate my previous major concern, that the model cannot adequately reconstruct testate amoebae inference with such a low count of tests.
Re: We believe that despite the low shell counts and species number in the zones TA1 and TA2 the boundary reflects changes in the shell preservation and a shift in the deposit types, therefore can be considered as representative.
Lines 269-293: The functional trait approach may not have yielded adequate results due to low count numbers. I think this section should be described using a qualitative rather than quantitative approach.
Re: The issue with the low per sample preservation of shells has been addressed in the previous replies to the reviewer. With this in mind, we specifically provided description of the trait characteristics per zone in the lines 304-313.
Figure 4. Zone 1 and Zone 2 show erratic trend. This may be due to low test count.
Re: Indeed, the trend might seem erratic, that might be related to a number of reasons that we discussed in the paper (e.g., to poor preservation of shells, the greater variation in the functional traits in sub-fossil communities as compared to the training set and linear modeling approach).
Discussion section: The section must be re-written to reflect the major challenge with test preservation issues. This should be noted at the start of the section. Reconstructed environmental conditions should be discussed cautiously. The authors should just focus on the testate amoebae autecology, which they have done but be critical of their inference models.
Re: We mention the issue of low preservation shells at the beginning of the section and rephrase the text to put more focus on the qualitative interpretations of surface wetness reconstruction. We limitations of the study are also stressed, especially at the bottom zone of the deposits (e.g., see the lines 361-363 of the original manuscript).
Comments on Figures:
Figure 1 Can you provide a better map with contours.
Re: Done as suggested.
Figure 2. Bottom x-axis (relative abundance %) not legible. Morphotypes and morphospecies names are hard to read. Can you make them vertical and larger? I don’t like the shaded lines; can you make these bars instead. Also, the samples with low testate amoebae counts (<50 sub-fossils) should be highlighted.
Re: The testate amoeba diagram has been revised. To make it readable we included only taxa which were observed in five or more samples or with the relative abundance greater than 3% per sample. That allowed us to increase the size the morphotypes. The width of the panels is now proportional to their abundance that allowed us to make the bottom x-axis legible. We have added a bar plot to the right-hand side of the diagram with the counts of testate amoebae per sample and marked the 50 shells with a red dotted line.
A figure with photomicrographs of the specimens should also be included in the manuscript.
Re: We have included a figure with the images of testate amoeba taxa as supplementary material Figure S1.
Reviewer 2 Report
Tsyganov et al. present a descriptive study of subfossil testate amoebae of an Indonesian peatland, applying species-based and trait-based approaches to reconstructing hydrological changes at multi-centennial timescales spanning the past ~13000 cal years. The manuscript was well-written, nicely organized, and easy to follow. Strengths of the work include the fact that there are few other tropical peatland hydrological reconstructions and this presents a nice case-study of the applicability of the approach. Furthermore it adds to the ongoing literature discussion regarding trait-based versus species-based approaches.
A weakness of the work include its lack of discussion of the paleoclimate history of the region to place the record into context, and the paper would be significantly improved if there were a deeper discussion of how the hydrological reconstructed related to broader (regional) development of ecosystems and climate over this time period. However, more importantly in my opinion and something that would give the work significantly more impact, would be if the hydrological reconstruction were compared with peat and carbon accumulation rates from the core. The authors begin the paper by highlighting the importance of tropical peatlands in C storage, mention in the methods that LOI and TOC were measured, but then never compare the hydrological history or testate amoebae changes with estimates of peat and carbon accumulation. I was surprised that they didn't make this comparison.
A few other minor comments:
Add the diameter of the core / coring device.
Add citation for a D-chamber corer.
What material was radiocarbon dated? Bulk peat or macrofossils?
A lot of the discussion uses the terms "stabliity" and "hydrological regime." I think the authors need to be clearer about what they are referring to with these terms and how "stability" in particular is being assessed. These data can only speak to variability at the timescales that they were collected, and given the coarse sampling intervals in one portion of the core (10cm) and the lower accumulation rate in the others, they really are only discussing changes at multi-centennial to millennial timescales. This needs to be explicitly stated, and some the mechanisms proposed (e.g., river flooding) need to be considered at these same timescales.
Author Response
Tsyganov et al. present a descriptive study of subfossil testate amoebae of an Indonesian peatland, applying species-based and trait-based approaches to reconstructing hydrological changes at multi-centennial timescales spanning the past ~13000 cal years. The manuscript was well-written, nicely organized, and easy to follow. Strengths of the work include the fact that there are few other tropical peatland hydrological reconstructions and this presents a nice case-study of the applicability of the approach. Furthermore it adds to the ongoing literature discussion regarding trait-based versus species-based approaches.
Re: We appreciate your positive feedback.
A weakness of the work includes its lack of discussion of the paleoclimate history of the region to place the record into context, and the paper would be significantly improved if there were a deeper discussion of how the hydrological reconstructed related to broader (regional) development of ecosystems and climate over this time period. However, more importantly in my opinion and something that would give the work significantly more impact, would be if the hydrological reconstruction were compared with peat and carbon accumulation rates from the core. The authors begin the paper by highlighting the importance of tropical peatlands in C storage, mention in the methods that LOI and TOC were measured, but then never compare the hydrological history or testate amoebae changes with estimates of peat and carbon accumulation. I was surprised that they didn't make this comparison.
Re: We agree that placing the record into a regional context of paleoclimatic history would improve the discussion; however testate amoebae indicate local changes in hydrological regime which difficult to extrapolate to a broader scale. Nonetheless, we discussed in the context of the mire development and regional changes in climate and human activity. The broader interpretation was provided elsewhere (see Hapsari et al. 2017, 2018). Regarding the carbon accumulation, the introduction of the submitted manuscript put too much focus on the carbon storage in tropical peatlands whereas our main goal was to describe the species composition of sub-fossil testate amoebae in peat deposits during the Holocene and reconstruct the hydrological regime. Therefore, we revised the text to put information about carbon storage only in the relation to the hydrological regime so that we removed the sentence “Recent studies indicate that the tropical 49 peatlands cover 23-30% of the total peatland area in the world, i.e., 90 to 170 Mha [2–4]” and corrected the last sentence in the paragraph as “However, despite the fact the hydrological regime is one of the main factors affecting peatlands, which also determines the storage and flux of carbon in these ecosystems, it has been rarely reconstructed in paleoecological studies on tropical peatlands.” Moreover, qualitative comparisons of reconstructed water table depths and carbon accumulation was provided in Discussion section of the original manuscript (lines L355-363; L395-399).
A few other minor comments:
Add the diameter of the core / coring device.
Re: done as suggested. In reads now: “core (a diameter of 5 cm)”
Add citation for a D-chamber corer.
Re: we added following citation “Aaby, B.; Digerfeldt, G. Sampling Techniques for Lakes and Bogs. Handbook of holocene palaeoecology and palaeohydrology 1986, 181, 194.”
What material was radiocarbon dated? Bulk peat or macrofossils?
Re: We add this information to the text: “To determine the age of the deposits, eight samples of the peat core (including bulk sediment, organic bulk sediment, organic samples, charred particles and seed material) were collected for Accelerated Mass Spectrometer radiocarbon dating (AMS 14C) which was performed at the University of Erlangen (Germany) and Poznan Radiocarbon Laboratory (Poland)”
A lot of the discussion uses the terms "stabliity" and "hydrological regime." I think the authors need to be clearer about what they are referring to with these terms and how "stability" in particular is being assessed. These data can only speak to variability at the timescales that they were collected, and given the coarse sampling intervals in one portion of the core (10cm) and the lower accumulation rate in the others, they really are only discussing changes at multi-centennial to millennial timescales. This needs to be explicitly stated, and some the mechanisms proposed (e.g., river flooding) need to be considered at these same timescales.
Re: We agree with the comment and revised the text of the manuscript accordingly. Indeed, water table depth/surface wetness is just one of the many possible characteristics of hydrological regime, so we tried to avoid unnecessary generalization and used “variable WTD/surface wetness” instead “unstable hydrological regime”.
Reviewer 3 Report
Dear Editor,
I recommend that the manuscript be accepted for publication in the present form. The manuscript is well written, the narrative is very fluid with a good level of English. The authors have done robust work on the paleoenvironmental reconstruction of the Sungai Buluh peatland in Indonesia, the construction of the age-depth model it is right, and also the reconstruction of water table depth from the testate amoebae communities, and the interpretation of the diversity and ecology of the taxonomic/functional groups based on the knowledge of the current ecology of testate amoebae. The discussion/conclusions are consistent with the results obtained and the specialized literature on the subject. The exposition of the ideas and their scientific basis is very adequate, at the same time the text has great narrative fluidity.
----
Minor questions:
Line 112 (Figure 1): Image has low resolution and can be improved.
Lines 125-126: `The water table in the 125 peatland fluctuates from 0.0 to 0.7 m (with a mean value of 0.3 m) below the peat surface.´ This sentence requires a citation to support it (or is it an observation of the authors?? ... if so, how has it been measured?).
Lines 182-183: This cite is not in the references `(Simpson and Oksanen, 2021).´, and it is not in the correct format.
Line 363: There is an extra space before the dot `[17] included just few aquatic biotopes . The hydrological preferences of two dominant´
Line 443: Misspelling in the title of the section 4.5, replace `speces´ by `species´.
Author Response
Dear Editor,
I recommend that the manuscript be accepted for publication in the present form. The manuscript is well written, the narrative is very fluid with a good level of English. The authors have done robust work on the paleoenvironmental reconstruction of the Sungai Buluh peatland in Indonesia, the construction of the age-depth model it is right, and also the reconstruction of water table depth from the testate amoebae communities, and the interpretation of the diversity and ecology of the taxonomic/functional groups based on the knowledge of the current ecology of testate amoebae. The discussion/conclusions are consistent with the results obtained and the specialized literature on the subject. The exposition of the ideas and their scientific basis is very adequate, at the same time the text has great narrative fluidity.
Re: We appreciate your kind and positive feedback.
Minor questions:
Line 112 (Figure 1): Image has low resolution and can be improved.
Re: done
Lines 125-126: `The water table in the 125 peatland fluctuates from 0.0 to 0.7 m (with a mean value of 0.3 m) below the peat surface.´ This sentence requires a citation to support it (or is it an observation of the authors?? ... if so, how has it been measured?).
Re: We added the reference to Hapsari et al 2017.
Lines 182-183: This cite is not in the references `(Simpson and Oksanen, 2021).´, and it is not in the correct format.
Re: Changed and added to the references list.
Line 363: There is an extra space before the dot `[17] included just few aquatic biotopes . The hydrological preferences of two dominant´
Re: we deleted extra space.
Line 443: Misspelling in the title of the section 4.5, replace `speces´ by `species´.
Re: Thank you, we changed it.
Round 2
Reviewer 1 Report
The latest version of the manuscript has addressed my comments and is much improved. I am happy for the manuscript to be published in it's current form. It's a very interesting study and the hard work merits publication.
Reviewer 2 Report
The authors have addressed my major concerns and I look forward to seeing the paper published.